# Quantification of Phenotypic Variability of Lung Disease in Children with Cystic Fibrosis

**DOI:** 10.3390/genes12060803

**Published:** 2021-05-25

**Authors:** Mirjam Stahl, Eva Steinke, Marcus A. Mall

**Affiliations:** 1Department of Pediatric Respiratory Medicine, Immunology and Critical Care Medicine, Charité-Universitätsmedizin Berlin, Corporate Member of Freie Universität Berlin and Humboldt-Universität zu Berlin, 13353 Berlin, Germany; eva.steinke@charite.de (E.S.); marcus.mall@charite.de (M.A.M.); 2German Center for Lung Research (DZL), Associated Partner, 13353 Berlin, Germany

**Keywords:** cystic fibrosis, early lung disease, magnetic resonance imaging, multiple-breath washout, newborn screening, noninvasive monitoring, outcome measure

## Abstract

Cystic fibrosis (CF) lung disease has the greatest impact on the morbidity and mortality of patients suffering from this autosomal-recessive multiorgan disorder. Although CF is a monogenic disorder, considerable phenotypic variability of lung disease is observed in patients with CF, even in those carrying the same mutations in the cystic fibrosis transmembrane conductance regulator (*CFTR*) gene or *CFTR* mutations with comparable functional consequences. In most patients with CF, lung disease progresses from childhood to adulthood, but is already present in infants soon after birth. In addition to the *CFTR* genotype, the variability of early CF lung disease can be influenced by several factors, including modifier genes, age at diagnosis (following newborn screening vs. clinical symptoms) and environmental factors. The early onset of CF lung disease requires sensitive, noninvasive measures to detect and monitor changes in lung structure and function. In this context, we review recent progress with using multiple-breath washout (MBW) and lung magnetic resonance imaging (MRI) to detect and quantify CF lung disease from infancy to adulthood. Further, we discuss emerging data on the impact of variability of lung disease severity in the first years of life on long-term outcomes and the potential use of this information to improve personalized medicine for patients with CF.

## 1. Introduction

Cystic fibrosis (CF) is an autosomal-recessive multiorgan disease caused by mutations in the cystic fibrosis transmembrane conductance regulator (*CFTR*) gene on chromosome 7 [1,2,3,4]. To date, over 2000 mutations in the *CFTR* gene have been identified and divided into six classes according to the resulting defect [5,6,7]. Class I-III mutations were shown to entail rather severe multiorgan affection with pancreatic insufficiency, diabetes mellitus, liver disease and impaired lung function, while classes IV-VI usually led to a milder disease trajectory [8]. Therefore, *CFTR* mutations leading to a complete lack of functioning protein are in general considered as severe genotypes, whereas a genotype with a residual CFTR function is defined as mild. A spectrum of molecular mechanisms including deficient production, folding, trafficking, regulation, or increased turnover lead to impaired protein synthesis and/or function of the CFTR protein, which functions as a regulated anion channel [9]. Patients with CF have abnormal transport of anions (chloride and bicarbonate) and fluid across the epithelia, resulting in mucus dehydration and thickened, highly viscous secretions in the bronchi and paranasal sinuses, but also in the biliary tract, pancreas, intestines and reproductive system [9,10,11]. Despite the impaired CFTR function in several organ systems, CF lung disease has the greatest impact on the quality and expectancy of life of affected patients [12,13]. The altered viscoelastic properties of airway secretions impair mucociliary clearance, which in turn leads to mucus obstruction of the small and large airways. Recent data showed that mucus plugging promotes hypoxic epithelial necrosis that can drive sterile inflammation, resulting in structural lung damage even in the absence of infection [14,15,16]. Further, mucus serves as a nidus for chronic infection for pathogens such as *Pseudomonas aeruginosa* as an important trigger of chronic airway inflammation [10,17,18]. Emerging evidence from several observational studies in infants and preschool children with CF diagnosed by newborn screening (NBS) suggests that CF lung disease starts in the first months of life with bronchial dilatation, air trapping, mucus plugging, ventilation inhomogeneity, neutrophilic inflammation and abnormal lung microbiota in often clinically unimpaired children [19,20,21,22]. These findings indicate that early therapeutic interventions may offer an opportunity to prevent or at least delay disease progression leading to irreversible lung damage. However, to date, the understanding of the onset and early course of CF lung disease and influencing factors including early treatment in infants and preschool children with CF remains limited [19,20,23,24,25,26,27,28]. A number of cross-sectional and longitudinal studies have indicated a high variability in the severity of early CF lung disease [29,30,31,32,33,34,35], even in a pediatric cohort with CF of a single center under a comparable symptomatic standard of care (Figure 1) [31]. Several factors may influence the variability of early CF lung disease, including the underlying *CFTR* genotype. The variability in the pulmonary course of Phe508del-homozygous patients with CF suggests that modifier genes, epigenetic modifications, age and mode of diagnosis and environmental factors are important modifiers of disease severity and are, therefore, addressed hereafter [36]. The detection of the individual level of CF lung disease is crucial to predict the prognosis and adapt the intensity of treatment of the individual patient to avoid both under-treatment as well as an unnecessarily high treatment burden. Until recently, sensitive endpoints able to detect the individual level of early CF lung disease in children were missing. In this review, we discuss current knowledge on factors leading to the broad phenotypic spectrum of CF lung disease, approaches for the quantification of this variability of early lung disease severity and knowledge gaps that need to be addressed by future research to optimize personalized therapy for children with CF.

## 2. Causes of Variability of CF Lung Disease

Although CF is a monogenic disease, a considerable clinical phenotypic variability in both presentation and clinical course is observed even in patients with the same *CFTR* genotype [36,37]. In adolescents and adults with CF, lung disease severity is associated with a severe *CFTR* genotype (the association of two *CFTR* mutations, including homozygosity for Phe508del, leading to either a complete lack of functioning protein or to a highly unstable protein with minimal activity, is considered as a severe genotype; by contrast, a genotype leading to a residual CFTR function is defined as mild), pancreatic insufficiency, chronic infection with *Pseudomonas aeruginosa*, presence of CF-related diabetes, lung function impairment and dystrophy [38]. Pulmonary exacerbations (PEx) as a hallmark of CF lung disease severity are also more common in older patients with CF harboring these characteristics [38]. However, the situation in infants and preschool children with CF is less clear.

### 2.1. The Role of the Underlying CFTR Genotype

Identification of the *CFTR* gene enabled studies of the genotype–phenotype relationship in CF [2,3,4,39]. A recent analysis of clinical data of patients aged 6–82 years collected in the cftr2 database and of their CFTR function revealed a moderate correlation between the forced expiratory volume in one second (FEV1) as a surrogate marker for the extent of lung disease and CFTR function determined by the CF genotype and calculated based on CFTR transport in rat thyroid and bronchial epithelial cells [5,40]. Despite the good correlation between sweat chloride and the CF genotype, there was a substantial variability in the relationship between FEV1 and sweat chloride concentration compared to the genotype-associated function [40]. Comparable data for younger patients with CF are lacking due to the lack of sensitive lung function endpoints in this age group prior to the recent renaissance of multiple-breath washout (MBW) as a measure of ventilation homogeneity that can detect CF lung disease from infancy on [29,31]. A longitudinal follow-up using a chest X-ray score in 132 children with CF demonstrated the strongest increase/worsening in children with one Phe508del allele in combination with a class I–III mutation followed by subjects homozygous for Phe508del, with the most stable course in those with one Phe508del allele in combination with a class IV–VI mutation [41]. A cross-age observation is that the exocrine pancreatic function in patients with CF appears to be strongly determined by the *CFTR* genotype, with patients carrying Phe508del or other class I or II and III mutations are, as a rule, exocrine pancreatic insufficient [42]. In contrast, *CFTR* class IV–VI mutations that are associated with residual CFTR chloride channel function typically confer pancreatic sufficiency, and patients carrying these alternative alleles are diagnosed later, have lower sweat chloride values, milder respiratory disease and a better prognosis than patients with two alleles associated with pancreatic insufficiency [10,43,44]. Therefore, exocrine pancreatic function is often used as a proxy for residual CFTR function to circumvent too many genotype-defined groups due to numerous rare *CFTR* mutations. Overall, pulmonary phenotypes of infants and preschool children carrying different *CFTR* genotypes have not been widely compared with sensitive outcome measures yet.

### 2.2. Identification of Modifier Genes

Heritability studies have shown that lung disease severity is variable among twins and siblings with CF and traits being affected by genetic and nonheritable factors [36,45,46,47,48]. Estimates assume that about half of the genetically determined variability can be attributed to the *CFTR* genotype with modifier genes accounting for the other half of the variability [49]. Of 114 CF twin and sibling pairs homozygous for Phe508del that were included in the European twin and sibling study, monozygous twins showed more concordance in lung function than dizygous twins, indicating that inherited factors modulate lung disease severity in addition to the *CFTR* genotype [36]. In a cohort of 101 sibling pairs with CF, severe lung disease as defined by Schluchter et al. [50] showed concordance in only about 20% of pairs, while the pancreatic status was highly concordant in sibling pairs (95%) [48]. In recent years, several studies using new technologies, such as genome-wide association studies and next generation sequencing, have been utilized for the search of “modifier genes”. To date, several studies have investigated the role of such modifiers in CF patient cohorts without a specific focus on infancy or early childhood. A number of candidate genes coding for or associated with alternative ion channels (e.g., solute carrier family 26 member 9, SLC26A9), bronchoconstriction, CFTR interaction, infection susceptibility and inflammatory response (e.g., interleukin-1 receptor 1 (IL-1R1)), nitric oxide, mucins (e.g., mucin 5AC) and oxidative lung injury have been identified [51]. A genetic variant of transforming growth factor β1 (*TGF-β1*) was found by single-nucleotide polymorphism technology and sequenced in a replication study, demonstrating an association with a lower FEV1 and a higher lung disease severity level [52]. These results, however, could not be reproduced in genome-wide association studies, leaving the role of this variant open [53]. Recent evidence has suggested that the microRNA miR-145 mediates the TGF-β inhibition of CFTR synthesis and function in airway epithelia, while specific antagonists to miR-145 interrupted TGF-β signaling to restore Phe508del CFTR modulation [54]. This indicates that the antagonism of miR-145 could be a novel therapeutic target in CF. Another recent investigation of patients with CF and their parents of the European twin and sibling study showed that an informative microsatellite marker within intron 1 of IL-1R was associated with a survival advantage of patients with CF [55]. These data support that IL-1R plays a role in CF pathogenesis in agreement with other studies on the IL-1R signaling pathway and its role in neutrophilic airway inflammation [56,57]. Of note, this also suggests a potentially interesting interaction between different factors contributing to the variability in CF lung disease. Furthermore, the relevance of a genetic modifier may depend on factors such as age at diagnosis or environmental factors. In other words, a gene that is relevant as a modifier in patients that are diagnosed late and are already chronically infected with *Pseudomonas aeruginosa*, i.e., a modifier of host defense, may not be relevant in patients diagnosed by NBS and vice versa. In addition, with increasing life expectancy, in part due to early diagnosis by NBS, modifiers may become relevant in the future, which did not play a role in patients that died in early adulthood. However, many of the studies to date are hampered by heterogeneity in the used outcome measures, limited reproducibility and missing clinical impact.

### 2.3. Identification of Epigenetic Modifications

Epigenetic modifications may represent the missing link between nonheritable factors and phenotypic variation in CF. Recent studies have shown altered deoxyribonucleic acid (DNA) methylation in CF [58], which may be caused by the overproduction of reactive oxygen species (ROS), depletion of DNA methylation cofactors and/or susceptibility to acute and chronic bacterial infections [59,60,61]. The unique DNA methylation profile of a single patient is thought to modulate the phenotype. A direct effect of bacteria on DNA methylation has not been shown yet, but a link among bacterial infection, the stimulated and activated immune cells that produce ROS and subsequent changes in DNA methylation is presumed. A reduced innate immunity, potentially influenced by modifier genes or environmental factors, may facilitate bacterial proliferation with accompanying inflammation and, therefore, drives CF lung disease [62]. To date, there are no data on epigenetic factors influencing the course and variability in early CF lung disease.

### 2.4. Impact of Age at Diagnosis and Mode of Diagnosis on Variability of Early CF Lung Disease

Thus far, only limited information is available on the impact of age at diagnosis following CF NBS on the severity and course of early CF lung disease [29,63,64]. This is mainly due to the lack of contemporaneously clinically diagnosed children, as CF NBS is usually introduced state- or nationwide, leaving no children behind that are clinically diagnosed early in life and treated according to the same standard of care [20,32,35,65]. In addition, at the time of initial implementation of CF NBS in populations with a contemporaneous group of children diagnosed clinically, sensitive outcome measures for CF lung disease were not available at many specialized CF centers. This did not only hamper the assessment of the role of NBS, but also of each of the other possible risk/influencing factors of variability in early CF lung disease [66,67]. The advent of sensitive outcome measures now provides an opportunity to investigate the different factors in infants and preschool children in more detail in future studies. First results indicate an advantage of early diagnosis following NBS with an immediate start of therapy, but further studies are needed [63,68,69,70].

### 2.5. Role of Environmental Factors

Exposure to environmental bacteria and fungi inadvertently leads to intermittent or chronic infection of the airways with pro-inflammatory pathogens, including *Haemophilus influenzae*, *Staphylococcus aureus*, *Pseudomonas aeruginosa*, *Streptococcus pneumoniae* and *Aspergillus spp*. These pathogens contribute to the course of lung disease in many pediatric patients [17,71]. *Pseudomonas aeruginosa* is one of the most prominent bacteria known to accelerate CF lung disease progression [34,72,73]. Infection with these (or several other) pathogens can lead to the slow progression of lung disease and/or repeated PEx. The latter has a strong impact on the trajectory of CF lung disease and correlates with a worse long-term outcome as the restoration of lung function by antibiotic treatment is not consistently achieved [74,75,76]. It was estimated that about half of the FEV1 decline in patients with CF can be attributed to PEx, with frequent successively occurring PEx negatively affecting lung function [76]. These findings could be confirmed in children with CF following NBS, where (i) PEx frequency in infancy was associated with lower FEV1, and (ii) PEx leading to hospital admission could be linked to bronchiectasis in computed tomography (CT) examinations at five years of age [77]. In addition, several factors, such as viral infections and nutritional status, are thought to affect the time to first PEx as well as the PEx rate [78,79,80,81]. Furthermore, an increased PEx rate, as well as reduced lung function were linked to the degree of air pollution in the residential area [82], consistent with findings suggesting that secondhand smoke exposure and environmental factors, such as a warm climate, air pollution or ozone exposure, contribute to increased infection with bacteria and fungi known to negatively affect CF lung disease [83,84,85,86,87,88].

The exposure of human bronchial epithelial cells to cigarette smoke extract (CSE) leads to acquired CFTR dysfunction due to the CSE-induced reduction in CFTR gating, decreased CFTR open-channel probability and partial reduction in surface CFTR expression with chronic exposure [9,89,90,91,92]. In addition, the exposure of neonatal mice with CF-like lung disease to cigarette smoke was shown to cause increased inflammation and mucus accumulation in the airways [93]. Secondhand smoke exposure in human infancy and childhood not only leads to more frequent hospitalizations, higher prevalence of air trapping and a shift in respiratory pathogens, but also to the activation of inflammatory pathways [83,85]. In addition, it is associated with lower cross-sectional and longitudinal lung function compared to nonexposed children, and this effect can be amplified by variations in the *CFTR* gene and a CF modifier gene (*TGF-β1*), highlighting the interaction between different causes of variability [73,83,94].

Compelling evidence shows a linkage among bacteria, inflammation and lung function deterioration present in the pediatric age range that partly determines disease variability [95]. Pittman et al. [61] used IL-8 and neutrophil counts in bronchoalveolar lavage fluid (BALF) as markers of inflammation in 32 infants with CF. A comparison of microbial diversity and inflammation demonstrated that diversity was lower in younger subjects and in those receiving daily antibiotic prophylaxis with a correlation between reduced diversity and lower IL-8 concentrations and absolute neutrophil counts [61]. However, inflammatory processes propelled by neutrophils can occur even in the absence of pathogens and have served in the identification of neutrophil elastase (NE) as promotor of structural changes in the lung from infancy on [14,35].

## 3. Methods to Detect and Quantify Early CF Lung Disease

To date, the specific contribution of each of the abovementioned potential causes of variability has not been determined in infants and preschool children with CF, and only partially in older patients with CF. The optimization and individualization of treatment in infants and preschool children with CF requires knowledge at the individual level of lung disease of single patients. This requires sensitive outcome measures that are able not only to detect early CF lung disease, but also to reflect different levels of disease severity in this young age group, allowing the adaption of treatment intensity to the extent of lung disease [20,29,96] (Table 1 and Table 2). Combining the results of investigations with sensitive outcome measures, deep clinical phenotyping and analysis of the previously mentioned risk/influencing factors can help to understand early variability and the potential role of these modifying factors.

Traditionally, spirometry-derived FEV1 is used as an endpoint to rate CF lung disease severity in noninvasive monitoring and various clinical trials. However, several factors limit its use in infants and preschool children with CF. First, spirometry depends on forced breathing maneuvers that are difficult to perform in naturally less cooperative young children. Second, due to the regional heterogeneity of early pulmonary findings starting in the lung periphery, this measurement of central airflow obstruction is not sensitive to detect the onset and progression of early CF lung disease (Table 1 and Table 2). The same is true for alternative endpoints derived by the raised volume-rapid thoraco-abdominal compression method, as results from both tests remain normal in most infants and children with CF on current treatment regimens [97,98]. To overcome the limitations associated with spirometry in the quantitative assessment of lung disease severity in children with CF, recent studies have focused on MBW as a lung function outcome and chest CT and magnetic resonance imaging (MRI) as imaging outcomes.

### 3.1. Investigation of Early Lung Function in Children with CF

Multiple-breath washout (MBW) measures ventilation inhomogeneity without the need for patients’ cooperation and is, therefore, a relevant tool for lung function measurement, especially in children [29,31,99,100] (Table 1 and Table 2). The MBW-derived lung clearance index (LCI) was shown to detect ventilation inhomogeneity from early infancy and to be more sensitive than FEV1 in detecting structural lung disease and response to therapy in older children with CF [25,28,29,98,99,101,102,103,104,105,106,107]. In addition, mean LCI z-scores were highly variable between the different age groups in childhood (Figure 1A) [31]. While about 70% of infants had normal LCI z-scores, the proportion of those with elevated z-scores increased with age to up to over 90% in adolescence [31]. Variability was high in all age groups with LCI z-scores ranging from −0.9 to 16.0 (age 0–1 years), −2.2 to 26.9 (age 2–5 years), −0.8 to 29.1 (age 6–11 years) and 1.0 to 26.7 (age 12–21 years) (Figure 1A) [31]. Variability in LCI at an individual level within a group of investigated children of the same age that started in infancy and progressed during childhood has also been demonstrated by several other studies on CF lung disease [29,30,33,98,104,108,109,110,111,112,113,114,115,116,117,118]. In addition, LCI has been used as an endpoint in studies investigating the efficacy of mucolytic therapies with hypertonic saline or rhDNase in CF from infancy to adolescence [25,28,101,102,107]. In addition to its ability to detect treatment responses at a group level by significant mean changes, the LCI also illustrated the variable baseline level of lung disease of the participants as well as their heterogeneous treatment responses [25,28,101,102,107].

Comparison of the LCI with cross-sectional imaging by chest CT and MRI suggests that the LCI also has limitations in detecting lung disease, especially at a very young age. While the LCI did not consistently detect structural changes in CT in infants with CF, a positive correlation between the LCI and structural lung disease was detectable in children from preschool age with a good positive predictive value of >80% [30]. This age-dependent difference is consistent with a recent study [31] demonstrating a higher rate of false-normal LCI values in younger children with CF who show elevated MRI scores compared to older children. One hypothesis that may explain this finding is that airways that are completely obstructed with mucus may not contribute to LCI, and infants and toddlers with smaller airway diameters may be more susceptible to complete mucus occlusion. Recently, first results have underlined the sensitivity of MBW to detect the response to real-life CFTR modulator therapy [119].

MBW-derived LCI could be further established as a sensitive endpoint for the phenotyping of early lung disease with good feasibility throughout childhood to bridge the knowledge gaps identified above.

### 3.2. Quantification of Morphological Changes Due to CF Lung Disease

LCI as a measure of global ventilation inhomogeneity is not able to provide information on the nature and localization of the abnormality leading to inhomogeneous ventilation. Therefore, cross-sectional imaging by CT and MRI offers the opportunity to investigate the cause (e.g., mucus plugging, bronchial wall thickening and consolidation) and regional distributions of morphological changes in CF lung disease, enabling even more specific therapeutic interventions.

The first morphological findings of CF lung disease detected by cross-sectional imaging are bronchial dilatation, bronchial wall thickening, air trapping and mucus plugging [20,65,120,121]. Prospective studies have identified evidence of bronchiectasis by CT in about 20% of infants shortly after diagnosis and in 50 to 75% of children with CF by three to five years of age [34,35,65] (Table 1 and Table 2). Important predictors of early bronchiectasis included a history of meconium ileus at birth, severe genotype, respiratory symptoms at the time of the study and NE activity in BALF [35]. Tracking of preschool CT findings into school age was demonstrated for a cohort of 61 children with CF, but the individual course was highly variable with about one third of patients showing an increase in CT scores, while the rest remained stable or even showed a decrease, indicating an improvement [122].

Conventional proton MRI as radiation-free imaging method has been implemented as an additional and informative tool within the past 15 years [123]. A semi-quantitative scoring system [124] enabled the determination of CF lung disease after diagnosis in infancy and early preschool years. The score comprises a lobe-based evaluation of six parameters representing typical pathologies [124]. The presence of each finding is rated from 0 (absent) over 1 (<50% of the lobe involved) to 2 (≥50% of the lobe affected) points with a maximum total score of 72. This morphofunctional score has shown sensitivity to detect worsening during PEx and response to antibiotic therapy [31,120]. MRI has been proven to be similarly sensitive to detect changes in the lung structure as CT, despite the higher spatial resolution of CT [125] (Table 1 and Table 2). As mucosal obstruction entails hypoxic vasoconstriction, lung perfusion assessment with MRI offers additional information, especially for the evaluation of nonvisible small airways (see Section 3.3). Recently, Ultrashort Echo Time MRI (UTE MRI) has emerged as a rapidly feasible tool and facilitates “CT-like” resolution for morphology studies [126,127], and first approaches to contrast-free MRI examination are under way [128]. The presence of structural changes varies widely between children of the same age, investigated in stable clinical condition and under the same standard of care using proton MRI (Figure 1B,C) [31]. The morphology score summarizes several findings, of which bronchial wall thickening/bronchiectasis and mucus plugging are the most prevalent in children with CF. Less frequent morphological findings are consolidations, pleural findings, abscesses and sacculations [31,120,129]. In a previous cross-sectional study in clinically stable patients across the pediatric age range, mean MRI morphology scores increased with age during childhood years (Figure 1C) [31]. Individual MRI morphology scores ranged from 1 to 15 (age 0–1 years), 3 to 15 (age 2–5 years), 0 to 28 (age 6–11 years) and 0 to 28 (age 12–21 years) (Figure 1C) [31]. Of note, only three children (all six years of age or older) showed no structural changes at the time of investigation (Figure 1C) [31]. A comparable picture emerged if the MRI global score was analyzed, summarizing both abnormalities in lung morphology and perfusion (MRI perfusion score; see below) that ranged from 0 to 40 over all age groups (Figure 1B) [31].

### 3.3. Detection of Lung Perfusion Defects by MRI

The advantage of MRI is the visualization of functional pulmonary impairments in addition to morphological findings. Thus far, most studies have been performed using the application of intravenous contrast agents in patients undergoing MRI examinations. Mucosal obstruction of airways leads to a consecutive hypoxic vasoconstriction in the adjacent blood vessel. Perfusion defects in a region of the lung are, therefore, an indirect sign of airway mucus plugging [130] (Table 1 and Table 2). In particular, small airways with diameters below the resolution of MRI (and CT) are thought to play a crucial role in CF lung disease development. Even within a stable cohort of children with CF aged 0 to 21 years, the MRI perfusion score showed variable levels of functional impairment within the different age groups and the total cohort (Figure 1D) [31]. In addition, intra-individual variability was also shown at a group level in infant, preschool and school-age children with CF at the time of PEx and after antibiotic therapy [31,120].

Alternative MRI techniques rather focus on the inhalation of hyperpolarized gas, such as helium or xenon, which discloses information on regional ventilation heterogeneity by evaluation of signal distribution [131,132]. This method correlates well with structural alterations in CF lung disease and is sensitive to detect pulmonary functional impairments [133,134,135]. Compared to spirometry, hyperpolarized gas MRI is more sensitive to detect even mild CF lung disease, is feasible in children and may serve as an outcome parameter in clinical studies [136]. The availability and logistics of these gases may constitute difficulties and hamper extensive performance in the clinical context. Various other MRI methods and their combinations to enhance the informative output have been successfully implemented, but are not entirely covered in the context of this review [137].

Overall, both CT and MRI can be used for the investigation and follow-up of early CF lung disease. MRI offers radiation-free examination of not only structural but also functional impairments due to changes typical for CF lung disease, such as bronchial wall thickening, bronchiectasis, mucus obstruction and hypoxic vasoconstriction.

### 3.4. Evaluation of Pulmonary Infection and Inflammation

BAL remains the gold standard for the investigation of the lower airways concerning the detection of pathogens and inflammatory markers (Table 1 and Table 2). Studies indicated neutrophilic inflammation associated with impaired lung function and lung function decline to be associated with the detection of *Pseudomonas aeruginosa* and *Staphylococcus aureus* [95]. Furthermore, the level of inflammatory markers, such as IL-8, NE or neutrophil counts, correlated with the LCI, indicating a progression of ventilation inhomogeneity with more pronounced inflammation [104]. IL-8 levels ranged from 0 to 8000 pg/mL in a cohort of infants and young preschool children [104]. Neutrophil counts in BALF in 58 children with CF aged five years were also widely distributed from 0.5 to 100% [115]. As many studies to date have focused on the description of findings at a group level, data on individual levels of inflammatory markers are often lacking. Frequencies of abnormal findings hint at the variability of individuals’ results. For example, a repetitive annual BALF analysis in two cohorts of preschool children with CF revealed that the percentage of BALF samples with detectable levels of NE was between 20% and 25% at all time points, but IL-8 was detected in 95% of samples [35,138]. The neutrophil counts increased with age, as did the percentage of BALF samples with detectable bacteria, again highlighting variability not only between individual patients at different ages, but also between different parameters of inflammation [35,138]. Most studies on lung microbiome and infection in infants and preschool children to date have focused on the identification of patterns of similarities in specific age groups without reporting individual values [22,139,140,141]. At least a visual variability in the pathogen load of BALF can be estimated without detailed reported ranges from a study in 78 five-year-old children [142].

As mentioned above, it is thought that the mucus obstruction of small airways precedes airway inflammation and infection. In this context, the study on mucins in preschool BALF comparing children with CF to non-CF children demonstrated that not only are more mucins present in CF than in controls irrespective of infection, but that mucus flakes in CF also have a higher density [21]. While mucin levels correlated with inflammatory markers and were already present in children with CF without structural lung disease or bacterial infection at a group level, the individual levels of both mucins and inflammatory markers varied substantially between children in this study from 0 to 6000 µg mucins/mL, 10^4^ to 10^8^ neutrophils/mL and 10^1^ to 10^5^ pg IL-8/mL [21].

Despite the information on the inflammation and infection of the lower airways that can be gained by bronchoscopy with BAL, the performance requires general anesthesia in infants and preschool children with CF, limiting its use for frequent follow up. In addition, a study investigating the effect of a BAL-directed antibiotic therapy compared to the standard of care in young children with CF was unable to demonstrate an advantage of the BAL-based approach [143]. It is possible that an anti-inflammatory approach based on the levels of inflammatory markers shortly after diagnoses, especially NE as main risk factor for bronchiectasis at preschool age, could have an effect on early CF lung disease progression [144].

## 4. Consequences of Variability of Early CF Lung Disease and Ability of Quantification

Lung function, structural lung disease, inflammation and infection have been shown to be highly variable between infants and preschool children with CF. Several risk/influencing factors have been identified that impact the severity of early CF lung disease at a group level, but their effect on individual patients remains unknown. MBW, CT and MRI as well as bronchoscopy with BAL are, in general, feasible techniques to detect these different features of early CF lung disease. Currently, the translation of these techniques into clinical practice is the next step to be able to adapt therapies to the individual level of lung disease in infants and children with CF. Together with new therapeutic approaches, this holds promise to further improve life expectancy and quality by reducing treatment burden in those with mild CF lung disease and provides the opportunity to intensify treatment in those with rapid progression in early life [145,146].

It is likely that novel treatments will work best before irreversible lung damage has appeared. Further, these medications might have risks, which means that at least shortly after approval, caretakers and parents might be undecided or unwilling to start these in young children coping clinically very well with symptomatic, established treatment regimens. Due to a different drug metabolism compared to adult patients with CF and rapid alveolar growth in the first 18 to 24 months of life, this is a particularly vulnerable period where little is known about medication safety in this context. MBW and imaging techniques can help not only to monitor treatment benefits, but also to screen for any potential damaging effects.

Therefore, as a first step, studies are needed investigating in detail the variability of early CF lung disease to build initial treatment decisions. This would be best conducted in a cohort of children under a comparable standard of care. Such an approach would also help to determine the effect of a standardized treatment on CF lung disease at a group level in young children. To date, the standard of care for the majority of children with CF is symptomatic therapy based on mucolytic treatments with the availability of CFTR modulating drugs for only a small group of infants with gating mutations and preschool children homozygous for Phe508del [24,27,147,148]. First results of a longitudinal MRI study in infants and young preschool children with CF under standardized symptomatic therapy indicate that an early start of therapy in asymptomatic infants following early diagnosis by NBS has the potential to keep CF lung disease at a lower level throughout the first years of life compared to a start of therapy following clinical diagnosis [69]. However, lung disease detected by MRI progresses in children diagnosed clinically and those following NBS with a comparable rate of decline, indicating the unmet need for more effective therapies that are able to change the trajectory of early CF lung disease [69]. Studies investigating whether an individualized therapy (compared to standard of care) is the road to success, that is, slowing or preventing progression of early CF lung disease, are the second step to follow. LCI, mucus plugging, wall thickening and perfusion defects in MRI can be used as endpoints to answer these questions.

## 5. Summary and Outlook

Early CF lung disease is highly variable, even in children with the same *CFTR* genotype. Several potential influencing factors have been identified, but these cannot explain the variability completely yet. Therefore, a better understanding of the onset and longitudinal course of early CF lung disease is crucial to be able to identify the most important, potentially modifiable risk factors. The deep-phenotyping of patient cohorts is necessary to gain more insights into the trajectories of structural and functional lung impairments in the early CF years. Sensitive outcome measures, such as MBW and MRI, can help to tackle this task. Furthermore, these methods can be used to monitor treatment responses and tailor individual therapy in children with CF in the era of emerging effective CFTR-directed therapeutics [146]. In addition, the use of multiomics approaches enabling integrated studies of changes in the transcriptome, proteome, metabolome, epigenome, microbiome and exposome in clinically well-characterized patient cohorts and their relationship to the variability of early CF lung disease has the potential to improve our current understanding of the nature of this variability. For example, the identification of changes in inflammatory pathways in children with specific clinical features or a distinct microbiome pattern could help to better understand the course of their disease and possibly adapt treatment strategies to these findings in the future. In addition to the identification of genetic modifiers, the investigation of the exposome, i.e., the sum of the environmental risk factors, holds promise for the development of new preventive strategies based on modifiable environmental factors that are associated with worse or more favorable outcomes of lung disease. The backbone for such future studies will be deeply characterized, closely followed patient cohorts and well-stocked biobanks.

## Figures and Tables

**Figure 1 genes-12-00803-f001:**
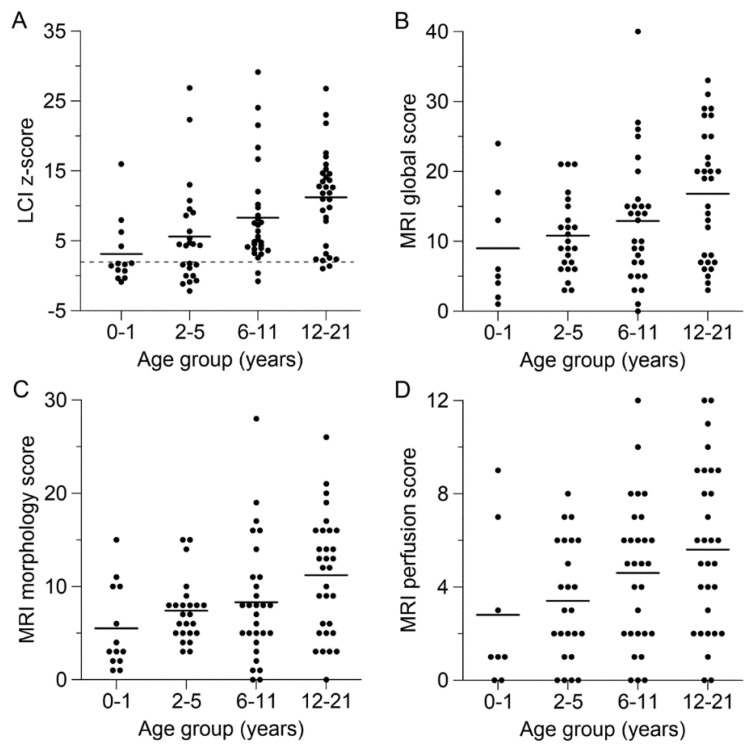
Variability of lung disease in children and adolescents with cystic fibrosis (CF) across the pediatric age range. (**A**–**D**) Level of lung disease in individual patients with CF determined by multiple-breath washout (MBW) and morpho-functional lung magnetic resonance imaging (MRI). Results are displayed for the lung clearance index (LCI) *z*-scores derived from age-adapted MBW using either sulfur hexafluoride (infants and toddlers) or nitrogen (children 4 years and older) as tracer gas (**A**), and the MRI global score (**B**), morphology score (**C**) and perfusion score (**D**). The solid lines indicate the mean of each age group, and the dashed line in panel A indicates the upper limit of normal for the LCI *z*-score (+1.96 SD). Perfusion studies were performed in 91 of 97 children with CF. Adapted from [31]. Adapted with permission of the American Thoracic Society. Copyright © 2021 American Thoracic Society. All rights reserved.

**Table 1 genes-12-00803-t001:** Overview on outcome measures for investigation of early CF lung disease discussed in this review.

Technique	Investigated Aspect of CF Lung Disease	Applicable Age Range	Advantages	Disadvantages
**Spirometry**	lung function	≥3 years	good availability	necessitates cooperation insensitive for mild changes
**MBW**	lung function	from infancy on (requiring sedation in some young patients)	performed in tidal breathing with minimal cooperation detects early ventilation inhomogeneities	only available at specialized centers harmonization between devices, tracer gases and protocols pending
**CT**	lung structure	from infancy on (requiring sedation in some young patients)	good availability short duration of performance high resolution images detecting early morphological changes	ionizing radiation (limiting repeatability) no information on lung function
**MRI**	lung structure lung function	from infancy on (requiring sedation in young patients)	sensitive to early CF lung disease can be repeated in short time (no radiation)	performed at specialized centers investigation takes longer than CT lower resolution than CT
**BAL**	infection inflammation	from infancy on (requiring anesthesia in young patients)	only way to properly investigate colonization of the lower airways and to measure inflammatory markers	invasive BAL-directed therapy has shown no advantage over standard therapy

Definitions of abbreviations: CF = cystic fibrosis; MBW = multiple-breath washout; CT = computed tomography; MRI = magnetic resonance imaging; BAL = bronchoalveolar lavage.

**Table 2 genes-12-00803-t002:** Overview of findings of early CF lung disease according to age group discussed in this review.

Finding	Infants and Toddlers	Preschoolers	School-Age Children	Adolescents
**Pulmonary function**				
Altered airway flow and resistance	Davies 2017 Hoo 2012 Kopp 2015 Lum 2007 Nguyen 2014 Pillarisetti 2011 Ramsey 2014	Gustafsson 2003 Gustafsson 2008 Ramsey 2014 Stahl 2017 Stanojevic 2017	Couch 2019 Fuchs 2008 Goss 2004 Gustafsson 2003 Gustafsson 2008 Kraemer 2005 McCague 2019 Ramsey 2014 Sanders 2015 Sanders 2014 Smith 2018 Stahl 2017 Svedberg 2018 Thomen 2017 Thomen 2020 Triphan 2020 Waters 2012 Willmering 2020 Zemanick 2015	Collins 2008 Couch 2019 Frey 2021 Fuchs 2008 Goss 2004 Graeber 2021 Gustafsson 2003 Gustafsson 2008 Kraemer 2005 McCague 2019 McKay 2005 Sanders 2014 Smith 2018 Stahl 2017 Svedberg 2018 Thomen 2017 Thomen 2020 Triphan 2020 Waters 2012 Willmering 2020
Hyperinflation	Davies 2017 Hoo 2012 Kieninger 2017 Nguyen 2014		Kraemer 2005	Kraemer 2005
Ventilation inhomogeneity	Belessis 2012 Davies 2017 Hoo 2012 Kieninger 2017 Lum 2007 Nguyen 2014 Ramsey 2016 Simpson 2015 Stahl 2018 Stahl 2014 Stahl 2017 Stahl 2019b Subbarao 2013 Triphan 2020 Warrier 2019	Aurora 2005 Aurora 2011 Belessis 2012 Downing 2016 Gustafsson 2003 Gustafsson 2008 McNamara 2019 Ramsey 2017 Ramsey 2016 Ratjen 2019 Singer 2013 Stahl 2018 Stahl 2014 Stahl 2020 Stahl 2017 Stanojevic 2017 Subbarao 2013 Triphan 2020 Warrier 2019	Amin 2010 Amin 2011 Aurora 2011 Couch 2019 Davies 2013 Fuchs 2008 Gustafsson 2003 Gustafsson 2008 Kraemer 2005 Ramsey 2016 Singer 2013 Smith 2018 Stahl 2017 Svedberg 2018 Triphan 2020 Willmering 2020	Amin 2010 Amin 2011 Couch 2019 Davies 2013 Fuchs 2008 Graeber 2021 Gustafsson 2003 Gustafsson 2008 Kraemer 2005 Singer 2013 Smith 2018 Stahl 2017 Svedberg 2018 Triphan 2020 Willmering 2020
**Imaging**				
Bronchial wall thickening	Eichinger 2012 Leutz-Schmidt 2018 Sly 2009 Stahl 2019a Stahl 2017 Stahl 2019b Stick 2009 Wielpütz 2014 Wielpütz 2018	Bouma 2020 Eichinger 2012 Leutz-Schmidt 2018 Stahl 2019a Stahl 2017 Stick 2009 Taylor 2020 Wielpütz 2014 Wielpütz 2018	Eichinger 2012 Stahl 2017 Thomen 2020 Willmering 2020	Eichinger 2012 Graeber 2021 Puderbach 2007 Stahl 2017 Thomen 2020 Willmering 2020
Bronchial dilatation/bronchiectasis	Eichinger 2012 Leutz-Schmidt 2018 Margaroli 2019 Mott 2012 Ramsey 2016 Sly 2009 Stahl 2019a Stahl 2017 Stahl 2019b Stick 2009 Warrier 2019 Wielpütz 2014 Wielpütz 2018	Bouma 2020 Eichinger 2012 Leutz-Schmidt 2018 Margaroli 2019 Mott 2012 Ramsey 2016 Stahl 2019a Stahl 2017 Stick 2009 Taylor 2020 Warrier 2019 Wielpütz 2014 Wielpütz 2018	Eichinger 2012 Bouma 2020 Gustafsson 2008 Ramsey 2016 Stahl 2017 Thomen 2020 Willmering 2020	Graeber 2021 Gustafsson 2008 Puderbach 2007 Stahl 2017 Thomen 2020 Willmering 2020
Mucus plugging	Eichinger 2012 Leutz-Schmidt 2018 Stahl 2019a Stahl 2017 Stahl 2019b Wielpütz 2014 Wielpütz 2018	Bouma 2020 Eichinger 2012 Leutz-Schmidt 2018 Stahl 2019a Stahl 2017 Taylor 2020 Wielpütz 2014 Wielpütz 2018	Eichinger 2012 Stahl 2017 Thomen 2020 Willmering 2020	Eichinger 2012 Graeber 2021 Puderbach 2007 Stahl 2017 Thomen 2020 Willmering 2020
Air trapping	Kopp 2015 Mott 2012 Ramsey 2016 Sly 2009 Stick 2009	Mott 2012 Ramsey 2016 Stick 2009 Taylor 2020	Gustafsson 2008 Ramsey 2016	Gustafsson 2008
Structural lung disease/abnormal CT or MRI score	Kopp 2015 Montgomery 2018 Muhlebach 2018 Ramsey 2014 Ramsey 2016 Rosenow 2019 Triphan 2020 Wainwright 2011	Esther 2019 Gustafsson 2008 Montgomery 2018 Muhlebach 2018 Ramsey 2014 Ramsey 2016 Rosenow 2019 Taylor 2020 Triphan 2020 Wainwright 2011	Gustafsson 2008 Ramsey 2014 Ramsey 2016 Triphan 2020	Gustafsson 2008 Triphan 2020
Perfusion deficits	Eichinger 2012 Leutz-Schmidt 2018 Stahl 2019a Stahl 2017 Triphan 2020 Wielpütz 2014 Wielpütz 2018	Eichinger 2012 Leutz-Schmidt 2018 Stahl 2019a Stahl 2017 Triphan 2020 Wielpütz 2014 Wielpütz 2018	Eichinger 2012 Stahl 2017 Triphan 2020	Eichinger 2012 Graeber 2021 Stahl 2017 Triphan 2020
VDP/VHI			Couch 2019 Smith 2018 Thomen 2017 Thomen 2020 Willmering 2020	Couch 2019 Smith 2018 Thomen 2017 Thomen 2020 Willmering 2020
**Inflammation/Infection**				
Inflammation	Belessis 2012 Deschamp 2019 Esther 2019 Kopp 2019 Linnane 2021 Margaroli 2018 Montgomery 2018 Mott 2012 Muhlebach 2018 Pillarisetti 2011 Pittman 2017 Ramsey 2014 Rosenow 2019 Sly 2009 Stick 2009	Belessis 2012 Esther 2019 Kopp 2019 Linnane 2021 Margaroli 2018 Montgomery 2018 Mott 2012 Muhlebach 2018 Ramsey 2017 Ramsey 2014 Rosenow 2019 Stick 2009	Kopp 2019 Ramsey 2014	Frey 2021 Graeber 2021
Infection/microbiome	Belessis 2012 Deschamp 2019 Esther 2019 Kopp 2015 Linnane 2021 Margaroli 2018 Montgomery 2018 Mott 2012 Muhlebach 2018 Pillarisetti 2011 Pittman 2017 Ramsey 2014 Rosenow 2019 Simpson 2015 Sly 2009 Stahl 2017 Stahl 2019b Stick 2009 Wainwright 2011 Warrier 2019 Wat 2008 Zemanick 2015	Belessis 2012 Esther 2019 Linnane 2021 Margaroli 2018 Montgomery 2018 Mott 2012 Muhlebach 2018 Ramsey 2017 Ramsey 2014 Rosenow 2019 Stahl 2017 Stanojevic 2017 Stick 2009 Taylor 2020 Wainwright 2011 Warrier 2019 Wat 2008 Zemanick 2015	Boutin 2017 Ramsey 2014 Sanders 2015 Stahl 2017 Wat 2008 Waters 2012 Zemanick 2015	Boutin 2015 Frey 2021 Graeber 2021 Stahl 2017 Wat 2008 Waters 2012
Increased mucus viscosity	Esther 2019	Esther 2019		
Pulmonary exacerbations	Byrnes 2013 Rosenfeld 2012 Stahl 2019a Stahl 2019b Wainwright 2011 Zemanick 2015	Byrnes 2013 Ratjen 2019 Rosenfeld 2012 Stahl 2019a Stanojevic 2017 Wainwright 2011 Zemanick 2015	Farhat 2013 Goss 2004 Zemanick 2015	Farhat 2013 Goeminne 2013 Goss 2004

Definitions of abbreviations: CT = computed tomography; MRI = magnetic resonance imaging; VDP = ventilation defect percent; VHI = ventilation heterogeneity index.

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
