# Peer review of "Quantification of Phenotypic Variability of Lung Disease in Children with Cystic Fibrosis"

_genes, 2021, doi:10.3390/genes12060803_

Round 1
Reviewer 1 Report
This review is well done, very comprehensive and focuses both on the usefulness of tests such as LCI and lung magnetic resonance, and on the long-term outcomes associated with the severity of lung disease in the first years of life.
In my opinion there are no particular limitations. The authors should better discuss the phenotypic variability in siblings with CF, citing some missing papers. It is also useful to mention (in section 2.1) the role of the CFTR genotype to predict pancreatic damage, both in pediatric and in adulthood (for example using PIP score).Author Response
Please see the attachment.

Reviewer 2 Report
Summary:
This review gives a very nice and detailed overview of where the field currently stands in detecting mild or subclinical disease in children with CF using state of the art methodologies such as Multiple-breath washout (MBW) measures and CT, MRI. These methods with increased sensitivity are necessary to allow better personalized medicine as soon as CF diagnosis has been confirmed, so treatment strategies can be tailored to the patient and adapted if necessary. Finally, current gaps and future directions are outlined, giving the reader a comprehensive overview of current and future directions in deep-phenotyping of CF patients, starting at a young age already.
Major comments:
Could the authors add a table giving an overview of all non-invasive methods (imaging, BAL) discussed in the review, and the pros and cons, such as: which lung disease parameters can be determined (e.g. lung structure, lung function, infection/inflammation), sensitivity, variability between repeated measurements, possibility to draw conclusions on individual level vs. group effects, age range for which method is proven useful, ...?
A second table would be nice to add which summarizes the main findings that could be drawn for each technique from the clinical studies described within the review, for children with CF at different age ranges. In this way, the readers would be able to grasp in a glance the key changes occurring early in life in children with CF.
Minor comments:
L40: defective chloride: add ‘and bicarbonate’ + accompanying reference
L54: “… CF lung disease starts in the first months of life in often clinically unimpaired children (Sly et al., 2009; Grasemann 55 and Ratjen, 2013).”
Could the authors please specify in a bit more detail what is meant with CF lung disease (which parameters were found abnormal?), especially in the context of no apparent clinical symptoms?
L88: “Although CF is a monogenic disease, a considerable clinical phenotypic variability in both presentation and clinical course is observed (Mekus et al., 2000; McKone et al., 2006).”
Can possibly more recent references be added? Clinical course has likely evolved since 2006, especially also since the 2012 approval of Kalydeco for gating mutations and additional modulator therapies more recently.
L103: Can the authors briefly add a line also on the correlation of sweat chloride to genotype function (i.e. good correlation, McCague et al., 2019), since they next mention both FEV1 and sweat chloride showing a high variability in clinical presentation even with the same level of genotype function.
L181: “In addition, at the time of implementation of CF NBS, sensitive outcome measures for CF lung disease were not available at many specialized CF centers. This did not only hamper assessment of the role of NBS, but also of each of the other possible risk / influencing factors of variability in early CF lung disease (Farrell et al., 1997; Farrell et al., 2001).”
Could the authors add more recent references for this statement? Newborn screening for CF is a rather recent event in many countries, so the outcome measures to identify early lung disease have likely evolved since then.
L222: Can a bit more experimental evidence shortly be added to explain the causal link between cigarette smoke and its effect on CFTR function? See for example: Am J Respir Cell Mol Biol, 2017 Jan;56(1):99-108. doi: 10.1165/rcmb.2016-0226OC. The same for a potential role of variations in TGF-beta and the link with increased CF severity?
Round 2
Reviewer 1 Report
I thank the authors for the changes made.
Author Response
We thank the reviewer for the kind feedback.